# Specific Nucleic AcId Ligation for the detection of Schistosomes: SNAILS

Alexander James Webb[1‡], Fiona Allan[2‡], Richard J. R. Kelwick[1‡], Feleke Zewge Beshah[3], Safari Methusela Kinung'hi[4], Michael R. Templeton[5], Aidan Mark Emery[2]*, Paul S. Freemont[1,6,7]*

**1** Section of Structural and Synthetic biology, Department of Infectious Disease, Imperial College London, London, United Kingdom, **2** Department of Life Sciences, Natural History Museum, London, United Kingdom, **3** College of Natural and Computational Sciences, Addis Ababa University, Arat Kilo, Addis Ababa, Ethiopia, **4** National Institute of Medical Research (NIMR), Mwanza Center, Mwanza, Tanzania, **5** Department of Civil and Environmental Engineering, Imperial College London, London, United Kingdom, **6** The London Biofoundry, Imperial College Translation and Innovation Hub, White City Campus, London, United Kingdom, **7** UK Dementia Research Institute Care Research and Technology Centre, Imperial College London, Hammersmith Campus, London, United Kingdom

‡ These authors joint first authorship on this work.
* a.emery@nhm.ac.uk (AME); p.freemont@imperial.ac.uk (PSF)

**Data Availability Statement:** All relevant data are within the paper and its Supporting Information files.

## Abstract

Schistosomiasis, also known as bilharzia or snail fever, is a debilitating neglected tropical disease (NTD), caused by parasitic trematode flatworms of the genus *Schistosoma*, that has an annual mortality rate of 280,000 people in sub-Saharan Africa alone. Schistosomiasis is transmitted via contact with water bodies that are home to the intermediate host snail which shed the infective cercariae into the water. Schistosome lifecycles are complex, and while not all schistosome species cause human disease, endemic regions also typically feature animal-infecting schistosomes that can have broader economic and/or food security implications. Therefore, the development of species-specific *Schistosoma* detection technologies may help to inform evidence-based local environmental, food security and health systems policy making. Crucially, schistosomiasis disproportionally affects low- and middle-income (LMIC) countries and for that reason, environmental screening of water bodies for schistosomes may aid with the targeting of water, sanitation, and hygiene (WASH) interventions and preventive chemotherapy to regions at highest risk of schistosomiasis transmission, and to monitor the effectiveness of such interventions at reducing the risk over time. To this end, we developed a DNA-based biosensor termed **S**pecific **N**ucleic **A**c**I**d **L**igation for the detection of **S**chistosomes or 'SNAILS'. Here we show that 'SNAILS' enables species-specific detection from genomic DNA (gDNA) samples that were collected from the field in endemic areas.

## Author summary

Schistosomiasis is a neglected tropical disease, caused by the parasitic trematodes of the genus *Schistosoma*. Schistosomiasis is endemic to regions within Africa, Asia and South

**Funding:** AJW, FA, FZB, SMK, MRT, AME and PSF were funded by the UK Government's Global Challenges Research Fund (GCRF) through the Engineering and Physical Sciences Research Council (EPSRC) grant [EP/P028519/1] as part of the WISER project. https://gow.epsrc.ukri.org/NGBOViewGrant.aspx?GrantRef=EP/P028519/1 AJW and PSF received funding from the Imperial College London EPSRC Impact Acceleration Account [EP/R511547/1] https://www.imperial.ac.uk/research-and-innovation/research-office/funder-information/funding-opportunities/internal-funding-opportunities/iaa/epsrc/. RJRK and PSF received funding from Biotechnology and Biological Sciences Research Council (BBSRC) grant [BB/T017147] and [BB/W012987/1]. https://bbsrc.ukri.org/research/grants/grants/PersonDetails.aspx?Personid=-27428 The funders had no role in study design, data collection and analysis, decision to publish, or preparation of the manuscript.

**Competing interests:** The authors have declared that no competing interests exist.

America with at least 250 million people infected and a further 779 million at risk of infection. The lifecycle of schistosomes are complex and involve specific freshwater intermediate snail hosts which shed infective cercariae into the waterbodies they inhabit. Schistosomiasis is subsequently transmitted to humans or animals that contact cercariae contaminated water. In Africa, human disease is largely caused by *Schistosoma mansoni* and *Schistosoma haematobium*. However, endemic regions also typically feature animal-infecting schistosomes that can have broader economic and/or food security implications. Therefore, the development of species-specific *Schistosoma* detection technologies may help to inform local environmental, food security and health programmes. To this end, we re-purposed a nucleic acid detection technology to enable the detection of different schistosome species. Our DNA-biosensor, abbreviated as 'SNAILS', employs carefully designed probes that recognise species-specific DNA sequences, coupled with enzymatic amplification steps, and a fluorescent signal-dye to indicate a positive detection. 'SNAILS' successfully differentiates between *S. mansoni* and *S. haematobium* samples and could conceivably be employed within future global health programmes.

## Introduction

Schistosomiasis is a neglected tropical disease (NTD), caused by parasitic flatworm trematodes of the genus *Schistosoma* [1]. Worldwide, over 250 million people are infected with a further 779 million people at risk of infection, and an annual mortality thought to be 280,000 people in sub-Saharan Africa alone [2–4]. As such, it continues to be a considerable cause of concern to global health, particularly in low- and middle-income (LMIC) countries. Indeed, it has been described as second only to malaria worldwide in terms of human suffering caused by tropical disease [1]. As well as Africa, schistosomes are found in the Arabian Peninsula, South America, China, the Philippines and Indonesia, and more recently southern Europe (Corsica) [5,6]. 'Schistosomiasis' is something of a blanket term covering a series of related but distinct diseases, caused by several different species of schistosome, and not all species of schistosome cause human disease. Whilst causing considerable damage to human health, schistosomiasis also causes considerable harm to cattle and other livestock, thus negatively impacting on local economies and food security. However, different combinations of human, veterinary and wildlife infective schistosomes can co-exist, giving rise to a highly complex and dynamic picture, thereby complicating their detection [7]. Therefore, the ability to rapidly identify species of schistosomes relevant to human health, as well as differentiate them from related sympatric schistosome species is highly desirable. In Africa, the human disease can be broadly differentiated into intestinal schistosomiasis (predominantly caused by *Schistosoma mansoni*) and urogenital schistosomiasis (caused by *Schistosoma haematobium*) [3]. It is important to also be able to differentiate between these two species, particularly as there is data suggesting that *S. mansoni* and *S. haematobium* require different treatment regimens [8].

The main control intervention for schistosomiasis for the past 20 years has been preventive chemotherapy via the mass drug administration (MDA) of praziquantel. Other interventions within the control toolbox have recently become more popular for example, water, sanitation and hygiene (WASH) interventions (including behavioural change) and snail control to reduce schistosomiasis transmission [9]. However, this disease disproportionally affects LMIC countries, which may have inadequate infrastructure for WASH provisions and other preventive measures [4,9]. The presence, and persistence, of this disease has a direct impact on the realisation and application of some of the UN Sustainable Development Goals; goal 3 "Good

health and well-being: ensure healthy lives and promote wellbeing for all at all ages", and goal 6 "Clean water and sanitation: clean accessible water for all" [10].

As the main control intervention for schistosomiasis has been preventive chemotherapy via MDA, most diagnostics available have focused on the detection of the parasites in human and to a lesser degree, veterinary relevant animal populations. However, human or veterinary parasitological surveys can take time and to progress towards elimination it may desirable to also detect transmission in less traditional ways, such as using environmental DNA (eDNA) in water or soil, in order to gain a more complete perspective on schistosomiasis transmission. This study, as part of the Water Infrastructure for Schistosomiasis-Endemic Regions (WISER) project, has therefore, centred on a water and environmental monitoring approach for the detection of schistosomes. Such environmental surveillance data may ultimately inform local environmental, food security and health-based policy making [11] and aid in the realisation of the UN Sustainable Development Goals. Standard snail surveys (collecting snails and looking for infection) [12] and water sampling (looking for free swimming cercariae) [13], which can also include the use of natural oil laced traps [14], have been undertaken. These techniques, though, are labour intensive and lack specificity, as many species of cercariae are morphologically similar. The use of quantifiable mouse bioassays, where the mice are in a cage and dipped into water, so that their tails, paws and portions of the lower abdomen are wetted in either *in situ* water samples or pre-collected water samples, have also been employed [15]. The drawback of mouse bioassays is the considerable time lag between infection and observation of possible infection, which requires the killing and dissection of the mice and then the counting of any adult schistosomes present. In regards to snail surveys, if snails collected *in situ* show no visible signs of infection i.e. shedding cercariae, they can be further analysed for prepatency (the presence of *Schistosoma* sporocysts), whereby the snails are crushed and analysed microscopically [16] and/or via molecular techniques such as using PCR-based methods [17–19] or loop-mediated isothermal amplification (LAMP) assays [20,21]. Furthermore, the detection of *S. mansoni* via the proteolytic activity of cercarial elastase, an enzyme it uses to invade thedefinitive host [22,23], on whole-cell bioreporters (WCBs) [24] or functionalised polyhydroxyalkanoates (PHAs)-based bioplastic beads [25] has also been tested. However, the use of WCBs within global health settings is challenging with many complex practical, societal, cultural, regulatory and data protection concerns needing to be addressed [26].

Recently, molecular approaches have emerged that involve the analysis of parasite-specific eDNA using water or soil samples [27,28]. These collection methods are relatively straightforward, and these samples can be analysed for the presence of schistosomes using a range of molecular techniques including polymerase chain reaction (PCR) [29,30], real-time PCR (RT-PCR) [31–33], multiplex RT-PCR [34], DNA sequencing [35], oligochromatographic dipsticks [33], LAMP assays [20,21] and a recombinase polymerase amplification fluorescence assay (RPA)[36]. These techniques such as RT-PCR, would, typically, also be followed by a DNA sequencing step for validation and to enable species level identification. Although DNA sequencing is a useful tool, it is however, expensive, and furthermore, the probes used for techniques such as RT-PCR may not be able to differentiate very small changes in target DNA sequences. More recent nucleic acid detection technologies, including SHERLOCK [37] and INSPECTR [38], have undergone accelerated development cycles towards viral diagnostic applications. However, to the best of our knowledge these technologies have not yet been tested for the detection of *Schistosoma* [39]. Therefore, we developed and validated a species-specific DNA-based biosensor for detecting schistosomes via small changes in DNA sequences over a short target length (22 bases). We termed our novel DNA target-specific biosensor: **S**pecific **N**ucleic **A**c**I**d **L**igation for the detection of **S**chistosomes or 'SNAILS'. Here, we report that the probes designed for *S. mansoni* and *S. haematobium* are specific to their respective

species, using synthetic single stranded DNA (ssDNA) targets as well as adult worm genomic DNA (gDNA) from samples isolated in schistosomiasis-endemic areas.

## Results

### 'SNAILS' DNA-based biosensor development and target identification

The 'SNAILS' DNA ligation-based biosensor system described here is adapted from a light-up 'Spinach' aptamer-based biosensor for the detection of micro RNAs (miRNA) [40], which has recently been employed to detect SARS-CoV-2 in clinical samples [41]. Although multiple miRNAs have been identified in and are released by schistosomes at different stages of their life cycle [42–44], we decided that due to potential abundancy, lifespan and stability issues with miRNA samples taken from study sites, we would target single-stranded genomic DNA (ssDNA). As such the biosensor design was revised and optimised for ssDNA detection. The biosensor comprises two probes, A and B, that each recognise a different half (11 bases) of a 22-base target region (Fig 1A). Probe A is phosphorylated at the 5' end which also encompasses the 11-base target-complementary region, whilst the target-complementary region of probe B encompasses the 3' hydroxylated (-OH) end (S1A Table). If the two probes recognise a ssDNA target, they will anneal to their specific target sequence. Addition of T4 DNA ligase will only catalyse the formation of a covalent bond between the 5' and 3'ends of the two probes, if they specifically recognise and anneal to the target sequence. Probe A also encodes a 3' T7 promoter, which after successful ligation with Probe B and addition of T7 RNA polymerase will result in transcription of the 'Spinach' aptamer and fluorogen binding enabling a real-time fluorescence measurement (Fig 1A).

For this proof-of-concept study, we targeted the cytochrome c oxidase subunit 1 (*cox*1) mitochondrial gene of *S. mansoni* [45] and *S. haematobium* [46]. Representative *cox*1 sequences from *S. mansoni* and *S. haematobium*, along with those of the related species *Schistosoma bovis*, *Schistosoma guineensis*, *Schistosoma curassoni*, *Schistosoma rodhaini*, *Schistosoma japonicum*, *Schistosoma malayensis* and *Schistosoma mekongi* (S1B Table) were aligned using MUSCLE [47] to identify 22-base target regions that would differentiate between species (S1A). In order to decrease the likelihood of false positive or off-target detection, the probes were designed to maximise sequence divergence at the target region and in particular around the ligation junction (Figs 1A & 2A).

### Validation of the 'SNAILS' biosensor workflow, biosensor optimisation and specificity of the *Schistosoma* probes

To optimise the 'SNAILS' biosensor assay, a validation workflow was designed (Fig 1B). This workflow enabled the optimum target (Fig 1C; S1B; S1 Data) and probe concentrations (Fig 1D; S1C; S1 Data) to be identified as 50 nM of each. To test the specificity of the *S. mansoni*-specific probes, a range of single, double or multiple base changes were made to the *S. mansoni* wild-type target region (SM_WT), as indicated in red capitals (SM_B to SM_S; Fig 2A; S1A Table). The probes were specific for detecting the *S. mansoni* wild-type target (SM_WT), although there was some, albeit very low, signal detection for targets SM_C to SM_G, which only have 1-base difference from SM_WT (Fig 2A and 2B; S1D; S1 Data). However, when these *S. mansoni*-specific probes were tested against the corresponding target region of *S. haematobium* (SH_WT), *S. rodhaini* (SR_WT), *S. bovis* (SB_WT), *S. guineensis* (SG_WT), *S. curassoni* (SC_WT), *S. japonicum* (SJ_WT) and *S. mekongi* (Smek_WT), no signal was detected indicating that the probes were unable to recognise the corresponding target region of these related species (Fig 2A and 2B; S1D; S1 Data).

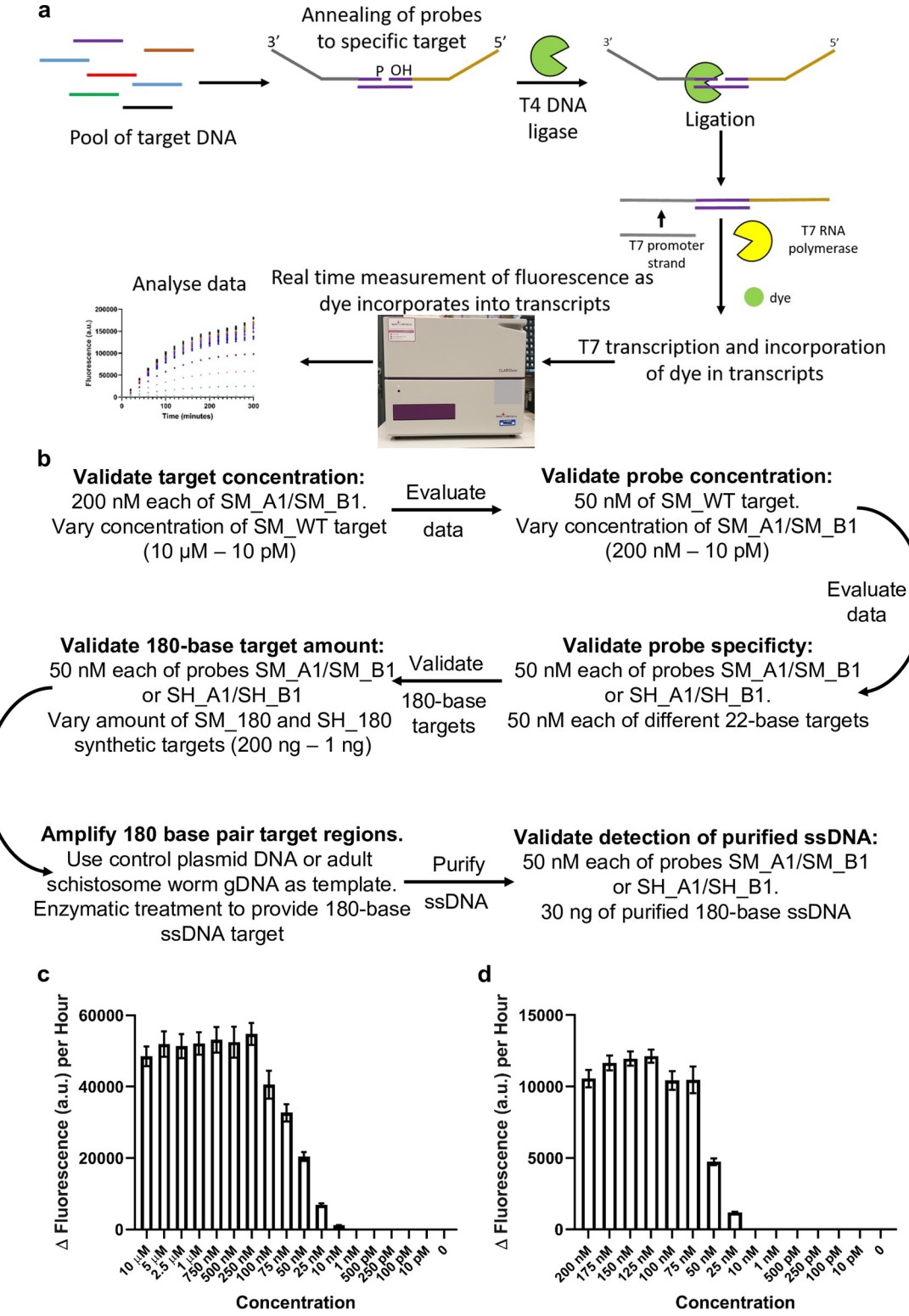

**Fig 1. "SNAILS" biosensor workflow and validation. a**, Workflow of the 'SNAILS' biosensors assay. **b**, Validation workflow followed for the 'SNAILS' assay. **c**, 'SNAILS' biosensor sensitivity. Sensitivity of *S. mansoni* probe set 1 (SM_A1/SM_B1) against a range of concentrations of the 22-base *S. mansoni* target (SM_WT, AJW791). Probe concentrations were set to 200 nM of each half probe. *n* = 12 (4 replicates per reaction, each reaction split into triplicate runs). **d**, Optimisation of probe concentration. Concentration of probes required to detect 50 nM of *S. mansoni* 22-base target (SM_WT, AJW791). *n* = 9 (3 replicates per reaction,

each reaction split into triplicate runs). Error bars denote standard error of the mean. The sequences of the probes and target are available in S1A Table.

Probes were also designed to recognise and detect the corresponding region of *cox*1 in *S. haematobium* (SH_A1 and SH_B1; S1A Table). The probes were specific for *S. haematobium* (Fig 3A and 3B; S1E; S1 Data), though there was an off-target response to the *S. curassoni* target sequence (Fig 3B). However, this output is markedly less and equated to only ~8.9% of the output observed for the *S. haematobium* target sequence. These data, therefore, validated the general 'SNAILS' biosensor assay.

## Validation of biosensor detection of purified target ssDNA

To test real-world parasitic gDNA and eDNA samples, the 'SNAILS' assay requires a PCR amplification step followed by an enzymatic step to produce a pool of ssDNA targets (Figs 1B & 4A). In this study, we chose to amplify a 180-base target region to validate the probes for *S. mansoni* and *S. haematobium* (S1A Fig). To assess whether the probes could recognise and bind to targets in 180-base long regions, the probes were tested against varying amounts (ng) of their respective synthetic 180-base long ssDNA targets (SM_180 and SH_180; S1A Table) and were able to bind resulting in detectable outputs (Fig 4B and 4C; S1F, S1G; S1 Data). We tested target amounts (ng) as real-world samples may have a range of nucleotide changes which could affect accurate molarity calculations. From the data (Fig 4B and 4C), 30 ng of target ssDNA appeared to be adequate for future 'SNAILS' assays.

To analyse the 'SNAILS' *Schistosoma* probes ability to recognise and anneal to PCR-derived 180-base long ssDNA samples (Figs 1B & 4A), we required *S. mansoni-* and *S. haematobium-*

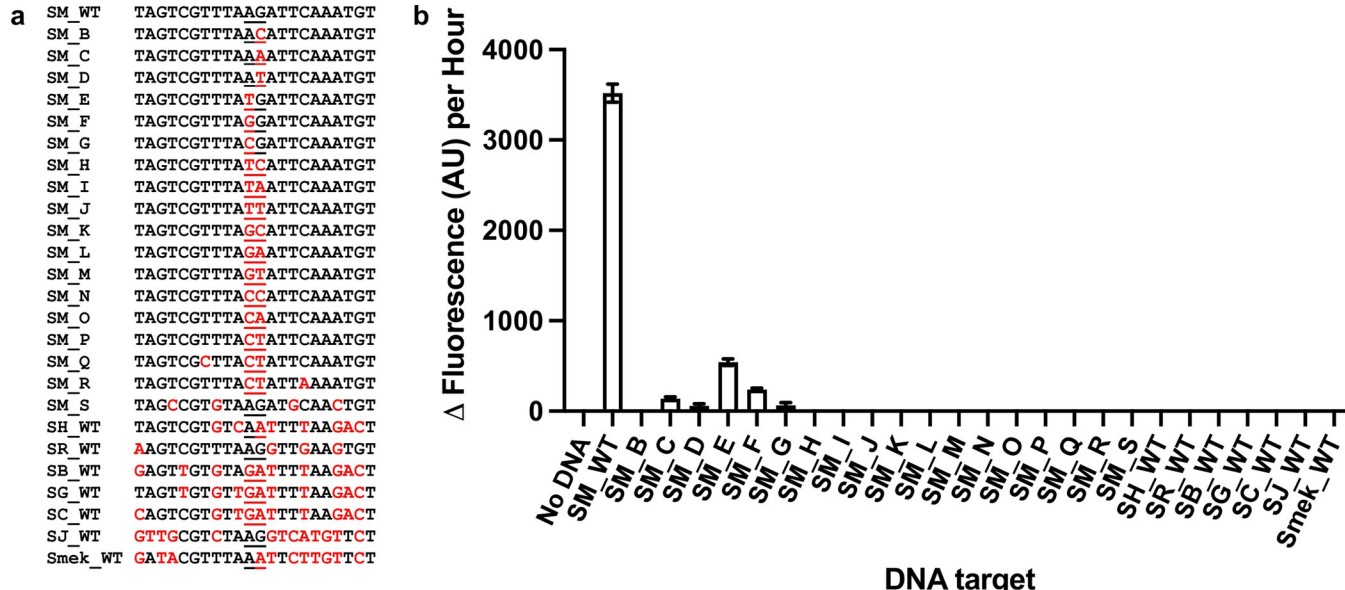

**Fig 2. Biosensor ligation junction and *S. mansoni* probe set specificity. a**, DNA targets tested. SM_WT is the target region probes SM_A1 and SM_B1 are designed to recognise. SM_B to SM_S have single, double or multiple base changes from that of SM_WT as indicated in red. The ligation junction for the probes is underlined. The corresponding target region for the other related *Schistosoma* species are indicated as follows: SR_WT (*Schistosoma rodhaini*), SH_WT (*Schistosoma haematobium*), SB_WT (*Schistosoma bovis*), SG_WT (*Schistosoma guineensis*), SC_WT (*Schistosoma curassoni*), SJ_WT (*Schistosoma japonicum*) and Smek_WT (*Schistosoma mekongi*). **b**, *S. mansoni* probes are specific for the *S. mansoni* wild-type target region. Both half probes and the target concentrations were tested at 50 nM. *n* = 9 (3 replicates per reaction, each reaction split into triplicate runs). Error bars denote standard error of the mean. The sequences of the probes are available in S1A Table.

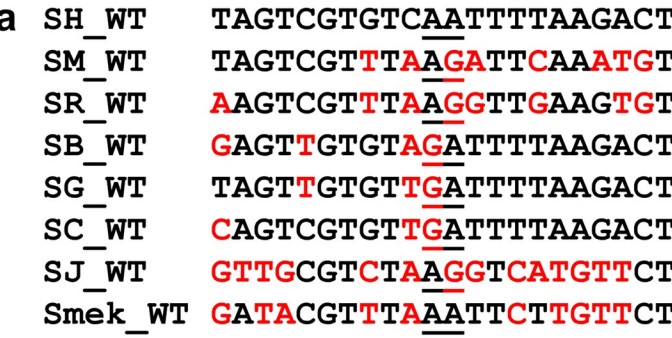
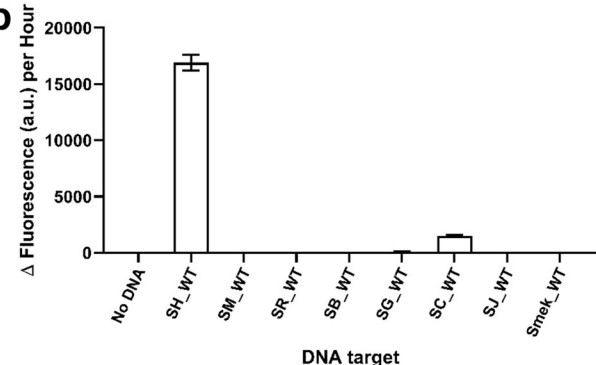

**Fig 3. *S. haematobium* probe set specificity. a**, DNA targets tested against *S. haematobium* probes. SH_WT is the target region that probes SH_A1 and SH_B1 are designed to recognise. The corresponding target region for the other related *Schistosoma* species are indicated as follows: SM_WT (*S. mansoni*), SR_WT (*S. rodhaini*), SB_WT (*S. bovis*), SG_WT (*S. guineensis*), SC_WT (*S. curassoni*), SJ_WT (*S. japonicum*) and Smek_WT (*S. mekongi*) with sequence variations indicated in red. The ligation junction for the probes is underlined. **b**, *S. haematobium* probes are specific for the *S. haematobium* wild-type target region. Both half probes and the target concentrations were tested at 50 nM. $n = 9$ (3 replicates per reaction, each reaction split into triplicate runs). Error bars denote standard error of the mean. The sequences of the probes are available in S1A Table.

derived control DNA. We, therefore, amplified *cox*1 from the gDNA of *S. mansoni* adult worm 1 (SM_AW1) and *S. haematobium* adult worm 1 (SH_AW1) and cloned the products into pCR-Blunt plasmids (pAJW250 and pAJW251; S1C Table). After sequence verification (S1D Table), we then amplified the 180-base target regions. The 3' primers are phosphorylated, enabling the removal of the 3' DNA strand via lambda exonuclease, leaving 180-base long ssDNA target fragments (Fig 1B & 4A). The *S. haematobium*-specific probes only detected the *S. haematobium* ssDNA samples, with no detection observed for the *S. mansoni* ssDNA samples (Fig 4E; S1J; S1 Data). However, the *S. mansoni*-specific probes compared to the respective positive control only, showed low detectable signals (an average of ~15.6% of that observed for the positive control) suggesting they were unable to sufficiently anneal to the PCR product (Fig 4D; S1I; S1 Data).

## Validation of alternative *S. mansoni*-specific targets and probes

Further analysis of the *S. mansoni* 180-base target region identified three new targets (S1K Fig). New probe pairs (S1A Table) were designed and validated against the 180-base ssDNA samples tested previously (Fig 4D & 4E), as well as the 180-base control oligonucleotides. *S. mansoni* probe sets 2, 3 and 5 successfully detected only *S. mansoni*-specific ssDNA samples, with no fluorescence output detected for the *S. haematobium*-specific samples (Fig 5A, 5B & 5C; S1L; S1 Data).

We then further analysed the probes for their specificity (Fig 5G, 5H & 5I; S1A Table). Probe sets 2 and 3 were able to recognise corresponding target regions of *S. rodhaini* (Fig 5D & 5E; S1M, S1N; S1 Data). However, probe set 5 was specific for *S. mansoni* (Fig 5F; S1O Fig; S1 Data). Therefore, *S. mansoni*-specific probe set 5 and *S. haematobium*-specific probe set 1 were further tested on species-specific ssDNA samples obtained from adult worm gDNA.

## Probes can differentiate ssDNA targets from adult schistosome worms

gDNA samples from *S. mansoni* and *S. haematobium* adult worms 1 and 2 were PCR amplified to generate 180-base target regions (Fig 6A & 6B; S1E Table). The target amplified from *S. haematobium* adult worm 1 gDNA (SH_AW1) had 2 base changes in the target region of *S. mansoni* probe set 5, one of which made it more similar to the *S. mansoni* sequence (Fig 6A; S1E

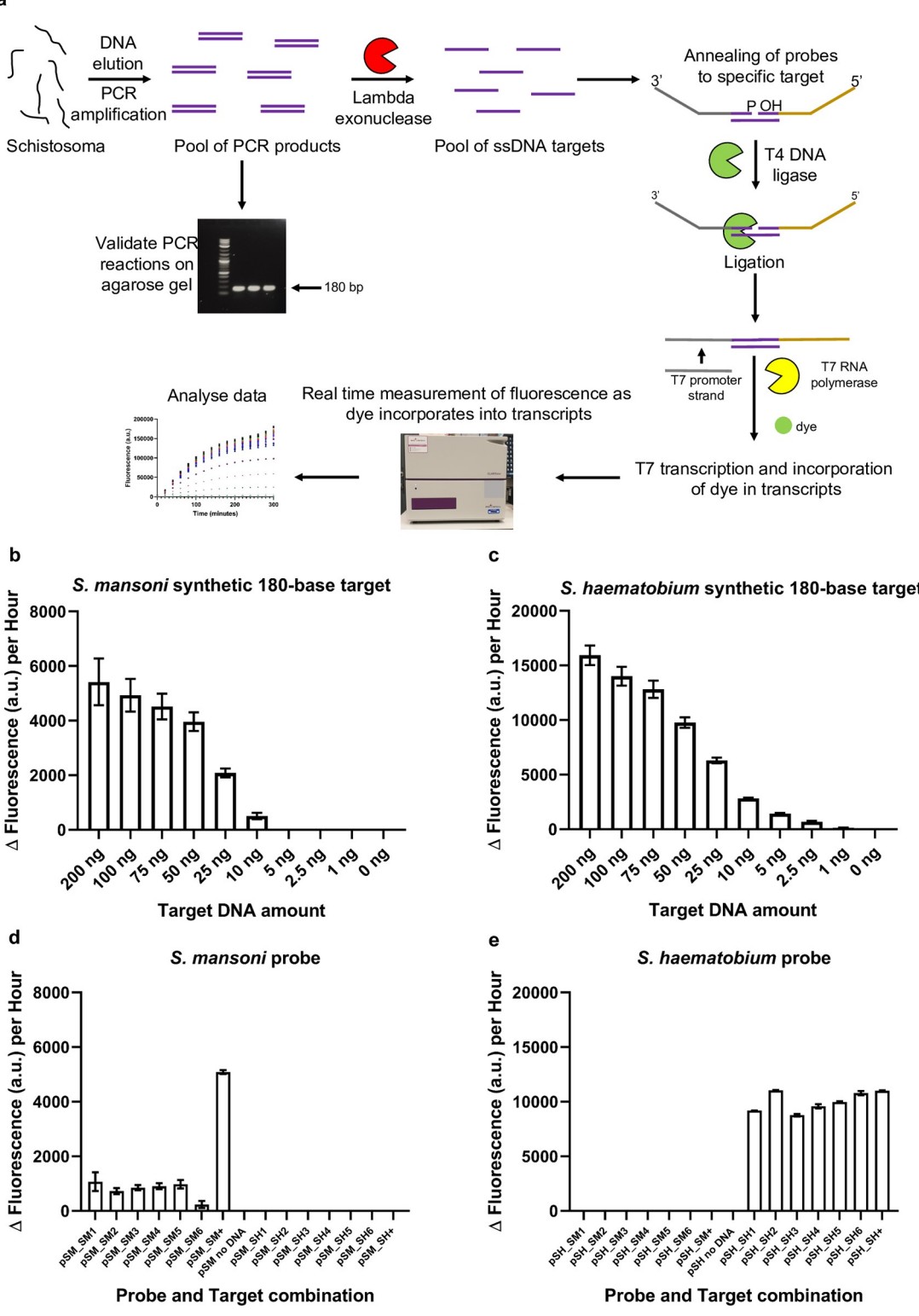

**Fig 4. SNAILS biosensor probe validation with oligonucleotide or PCR amplified ssDNA targets. a**, Schematic of an extended 'SNAILS' biosensor workflow showing the extra steps required to amplify the target and obtain purified ssDNA. Purified DNA (plasmid or adult worm gDNA) is amplified via PCR. PCR products are validated via agarose gel electrophoresis and then treated with lambda exonuclease to remove the phosphorylated 3' DNA strand. The resultant ssDNA is purified and then subsequently incubated with relevant probes in 'SNAILS' reactions, as described in Fig 1A. **b**, *S. mansoni* oligonucleotide

target. *S. mansoni* probes recognise and bind to a synthetic 180-base *S. mansoni* target. 50 nM of each probe half (SM_A1/ SM_B1) were incubated with a range of different amounts (ng) of the SM_180 target oligonucleotide (AJW874). **c**, *S. haematobium* oligonucleotide target. *S. haematobium* probes recognise and bind to a synthetic 180-base *S. haematobium* target. 50 nM of each probe half (SH_A1/SH_B1) were incubated with a range of different amounts (ng) of the SH_180 target oligonucleotide (AJW875). For both **b** & **c**, *n* = 9 (3 replicates per reaction, each reaction split into triplicate runs). **d**, *S. mansoni* PCR-derived target. **e**, *S. haematobium* PCR-derived target. **d** & **e**, *S. mansoni* and *S. haematobium* probes ability (respectively) to bind to ssDNA derived from plasmid DNA. The 180-base target regions were amplified from plasmids pAJW250 (*S. mansoni* insert DNA) and pAJW251 (*S. haematobium* insert DNA) and treated as described in Fig 4A to produce purified ssDNA. Six PCR reactions per plasmid sample were tested against both probe pairs. 50 nM of each probe half was incubated with 30 ng of target ssDNA. As controls, both probe pairs were also incubated against 30 ng of the synthetic targets SM_180 and SH_180. Reactions are identified as follows: pSM_SM1-6 (*S. mansoni* probes against *S. mansoni* plasmid derived ssDNA), pSM_SM+ (*S. mansoni* probes against *S. mansoni* synthetic target), pSM_SH1-6 (*S. mansoni* probes against *S. haematobium* plasmid derived ssDNA), pSM_SH+ (*S. mansoni* probes against *S. haematobium* synthetic target), pSH_SM1-6 (*S. haematobium* probes against *S. mansoni* plasmid derived ssDNA), pSH_SM+ (*S. haematobium* probes against *S. mansoni* synthetic target), pSH_SH1-6 (*S. haematobium* probes against *S. haematobium* plasmid derived ssDNA), pSH_SH+ (*S. haematobium* probes against *S. haematobium* synthetic target). *n* = 3 (each reaction split into triplicate runs). Error bars denote standard error of the mean. The sequences of the probes and targets are available in S1A Table.

Table). The sequences of the target regions for the other samples were as expected (Fig 6A & 6B; S1E Table). After generating ssDNA target samples, *S. mansoni*-specific probe set 5 and *S. haematobium*-specific probe set 1 were able to detect and differentiate between the target samples (Fig 6C & 6D; S1P, S1Q; S1 Data). There was some limited off-target detection for both probe sets (Fig 6C & 6D). However, we observed that the signal obtained for off-target probe detection was less than obtained for a true positive detection.

It has been established that there is natural intraspecies sequence variation in schistosomes [48]. We, therefore, analysed the target sequences of a range of available *S. mansoni* and *S. haematobium* isolates (Fig 6E & 6F; S1F and S1G Table). As well as the wild-type target sequence, that is the specific sequence the probes were designed to target, we identified 6 target sequence subsets for *S. mansoni* and 3 target sequence subsets for *S. haematobium* (Fig 6E & 6F; S1F and S1G Table). Next, we tested the ability of the *S. mansoni*-specific probe set 5 and the *S. haematobium*-specific probe set 1 to detect 6 more *S. mansoni* gDNA samples isolated from disparate geographical locations. *S. mansoni*-specific probe set 5 was able to successfully detect 3 of these samples (worm SM_C: *S. mansoni* 2740 isolated from Kenya, worm SM_D: *S. mansoni* 3112 isolated from Senegal, and worm SM_E: *S. mansoni* 4209 isolated from Oman; Fig 6 G; S1R; S1 Data). *S. haematobium*-specific probe set 1 had an off-target response to *S. mansoni* worm SM_D isolated from Senegal (Fig 6 H; S1S; S1 Data), however this output signal equated to only ~9.1% of the output observed for *S. mansoni*-specific probe set 5 for the same target sample. The variations in the target sequences for these further *S. mansoni* worm gDNA samples are highlighted in S1E Table.

For the probes tested in this study (Fig 6C, 6D, 6G & 6H) we observed that a fluorescence cut-off value (Δ fluorescence of ~4500 a.u. per hour) as well as using the different species-specific probe sets in tandem, allowed differentiation of true positives and off-target positives, thereby enabling the SNAILS protocol to discriminate ssDNA targets from different *Schistosoma* species (S1P, S1Q, S1R & S1S; S1 Data).

## Discussion

Schistosomiasis is a neglected tropical disease that is endemic to regions within Africa, Asia and South America, and is of an increasing burden to global health in that at least 250 million people are infected and a further 779 million at risk of infection [2]. In Africa, human disease is largely caused by *S. mansoni* and *S. haematobium*. However, endemic regions also typically feature animal-infecting schistosomes that can have broader economic and/or food security implications [7]. Therefore, the development of species-specific *Schistosoma* detection

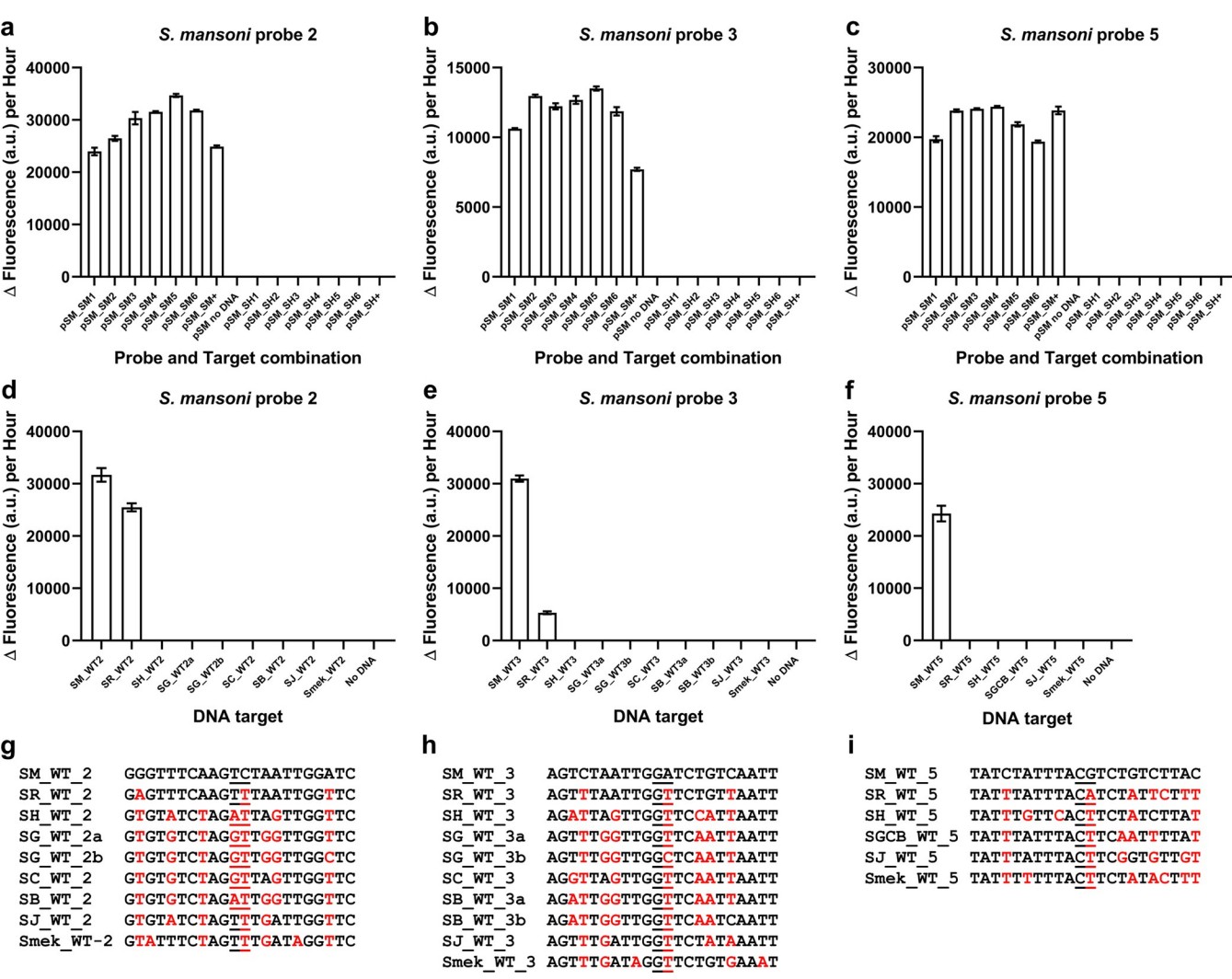

**Fig 5. Validation of re-iterated *S. mansoni*-specific probes. a, b & c**, Re-iterated *S. mansoni*-specific probe set designs 2, 3 and 5 respectively, only recognised and bound to *S. mansoni*-specific target ssDNA. 50 nM of each probe half was incubated with 30 ng of the purified ssDNA samples used in Fig 4D & 4E. Reactions are identified as follows: pSM_SM1-6 (*S. mansoni* probes against *S. mansoni* plasmid derived ssDNA), pSM_SM+ (*S. mansoni* probes against *S. mansoni* synthetic target), pSM_SH1-6 (*S. mansoni* probes against *S. haematobium* plasmid derived ssDNA), pSM_SH+ (*S. mansoni* probes against *S. haematobium* synthetic target). *n* = 3 (each reaction split into triplicate runs). **d, e & f**, *S. mansoni* probe sets 2, 3 and 5 respectively, were tested for their ability to recognise the corresponding 22-base target regions from other *Schistosomes*, which are indicated as follows: SM_WT2, SM_WT3 and SM_WT5 (*S. mansoni*), SR_WT2, SR_WT3 and SR_WT5 (*S. rodhaini*), SH_WT2, SH_WT3 and SH_WT5 (*S. haematobium*), SG_WT2a and SG_WT3a (*S. guineensis* AJ519523), SG_WT2b and SG_WT3b (*S. guineensis* AJ519522), SC_WT2 and SC_WT3 (*S. curassoni*), SB_WT2 (*S. bovis*), SB_WT3a (*S. bovis* MH647124), SB_WT3b (*S. bovis* FJ897160), SGCB_WT_5 (target sequence for *S. guineensis*, *S. curassoni*, and *S. bovis*), SJ_WT2, SJ_WT3 and SJ_WT5 (*S. japonicum*) and Smek_WT2, Smek_WT3 and Smek_WT5 (*S. mekongi*). Probe sets and the target concentrations were tested at 50 nM. *n* = 9 (3 replicates per reaction, each reaction split into triplicate runs). **a, b, c, d, e &f**, Error bars denote standard error of the mean. **g, h & i**, DNA targets tested against *S. mansoni* probes 2 (**g**), 3 (**h**) and 5 (**i**). Sequence variations with respect to the corresponding target sequence for the *S. mansoni* probes are indicated in red. The ligation junction for the probes is underlined. The sequences of the probes are available in S1A Table.

technologies may help to inform local environmental, food security and health programmes. However, as with other NTDs, historically there has been a significant underinvestment in commercial research and development programmes towards schistosomiasis diagnostics [49]. There are also limitations with currently used methodologies, several of which involve microscopic examinations of clinical or environmental *Schistosoma* samples. Whilst useful, microscopic examinations are labour intensive, highly subjective and do not enable the deeper insights that molecular approaches can conceivably provide. For example, the emergence of

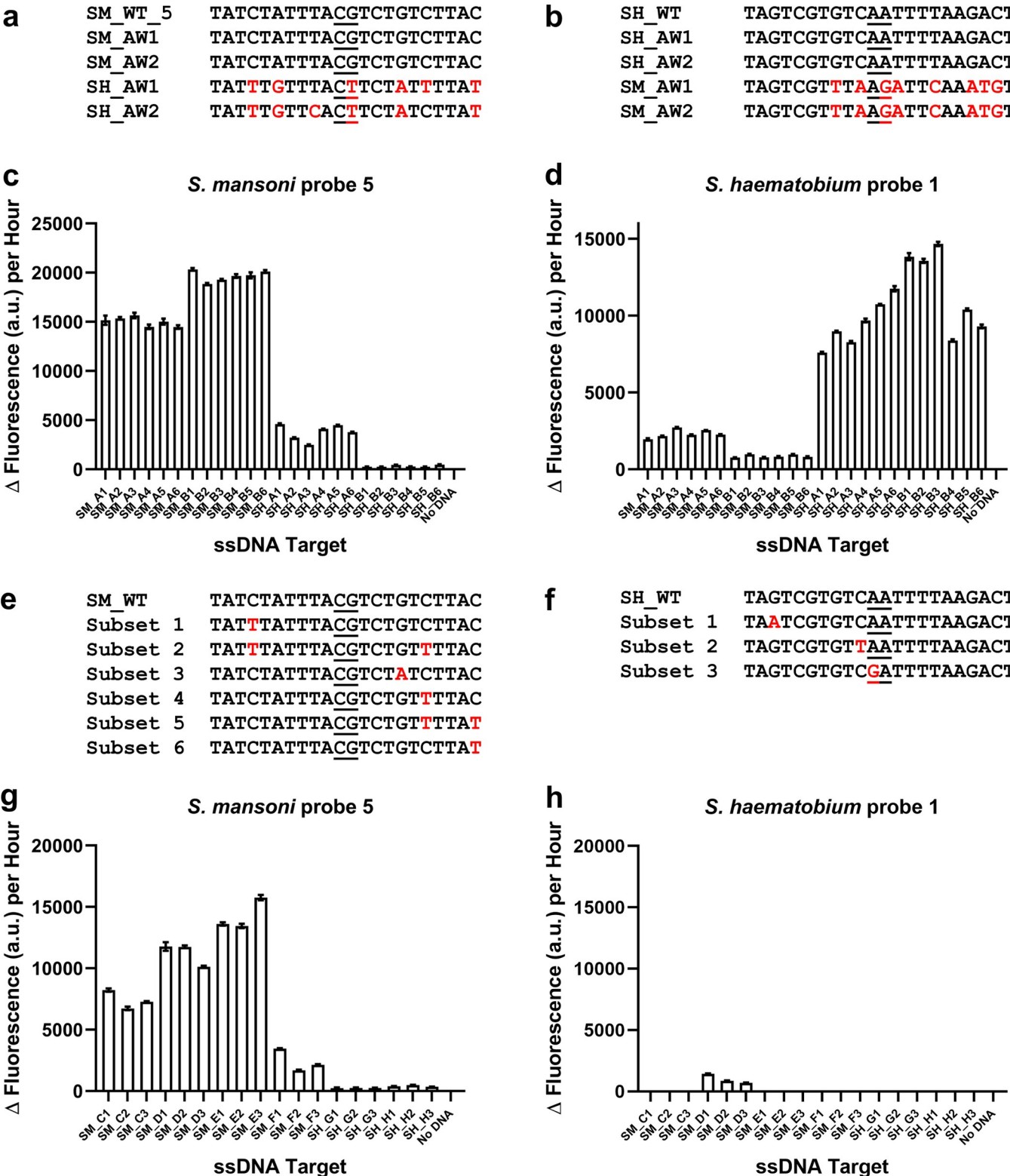

**Fig 6. Species specific detection of *S. mansoni* and *S. haematobium* from gDNA derived biological samples. a**, *S. mansoni* target region and **b**, *S. haematobium* target region and the corresponding regions from the four adult worm samples tested: *S. mansoni* adult worms 1 and 2 (SM_AW1 and SM_AW2 respectively) and *S. haematobium* adult worms 1 and 2 (SH_AW1 and SH_AW2 respectively). The differences between the target region and the corresponding regions of the adult worms are indicated in red. The probe ligation junctions are underlined. **c**, *S. mansoni* probe set 5 can differentiate between ssDNA derived

from adult worm gDNA. 50 nM of each probe half was incubated with 30 ng of purified ssDNA. Reactions are identified as follows: SM_A1-6 (*S. mansoni* adult worm 1 derived ssDNA samples), SM_B1-6 (*S. mansoni* adult worm 2 derived ssDNA samples), SH_A1-6 (*S. haematobium* adult worm 1 derived ssDNA samples), SH_B1-6 (*S. haematobium* adult worm 2 derived ssDNA samples). *n* = 3 (each reaction split into triplicate runs). **d**, *S. haematobium* probe set 1 can differentiate between ssDNA derived from adult worm gDNA. 50 nM of each probe half was incubated with 30 ng of purified ssDNA. Reactions are identified as follows: SM_A1-6 (*S. mansoni* adult worm 1 derived ssDNA samples), SM_B1-6 (*S. mansoni* adult worm 2 derived ssDNA samples), SH_A1-6 (*S. haematobium* adult worm 1 derived ssDNA samples), SH_B1-6 (*S. haematobium* adult worm 2 derived ssDNA samples). *n* = 3 (each reaction split into triplicate runs). **e**, intraspecies sequence variation of the *S. mansoni* target region. **f**, intraspecies sequence variation of the *S. haematobium* target region. **e & f**, nucleotide sequences used to identify sequence variation subsets are listed in S1F and S1G Table. **g**, *S. mansoni* probe set 5 detection of ssDNA derived from gDNA of *S. mansoni* isolated from disparate geographical locations. 50 nM of each probe half was incubated with 30 ng of purified ssDNA. Reactions are identified as follows: SM_C1-3 (*S. mansoni* 2740), SM_D1-3 (*S. mansoni* 3112), SM_E1-3 (*S. mansoni* 4209), SM_F1-3 (*S. mansoni* N1g274), SM_G1-3 (*S. mansoni* 4464–2), SM_H1-3 (*S. mansoni* 4464–4). **h**, *S. haematobium* probe set 1 detection of ssDNA derived from gDNA of *S. mansoni* isolated from disparate geographical locations. 50 nM of each probe half was incubated with 30 ng of purified ssDNA. Reactions are identified as follows: SM_C1-3 (*S. mansoni* 2740), SM_D1-3 (*S. mansoni* 3112), SM_E1-3 (*S. mansoni* 4209), SM_F1-3 (*S. mansoni* N1g274), SM_G1-3 (*S. mansoni* 4464–2), SM_H1-3 (*S. mansoni* 4464–4). *n* = 3 (each reaction split into triplicate runs). Error bars denote standard error of the mean. The sequences of the probes are available in S1A Table.

genomic DNA sequence variations within drug resistant (e.g. OXA) [50,51] or hybrid *Schistosoma* species [52] may not clearly manifest themselves at the morphological level but have important implications for global health programmes and affected patients.

To this end, we re-purposed a nucleic acid detection technology to enable the detection of different schistosome species. Our proof-of-concept data shows that our probe designs can detect and differentiate between two human health relevant schistosomes, *S. mansoni* (probe set 5) and *S. haematobium* (probe set 1), as well as discriminate these schistosomes from other closely related species including *S. bovis*, *S. guineensis*, *S. curassoni*, *S. rodhaini*, *S. japonicum*, and *S. mekongi*. Importantly, this study demonstrates that the 'SNAILS' probes are amenable to rapid design iterations that lead to improved biosensor sensitivity and specificity—that might also enable further improvements in base sequence resolution detection. Whilst the optimisation of biosensor probes has been an important aspect of this study, it is conceivable that other aspects of the 'SNAILS' assay could be refined in future studies. For example, the volumes or concentrations of assay reagents could be further optimised and open access enzymes (e.g., T7 polymerase) [53,54] could also be tested to reduce costs even further. Likewise, additional fluorescent molecule-binding aptamers could be utilised to enable compatibility with additional laboratory or in-field equipment [54,55] or enable assay multiplexing (detection of several DNA sequences within the same assay well). The global availability of PCR equipment has also rapidly expanded during the last several years and could be exploited to improve the accessibility of 'SNAILS' biosensor assays in different geographical locations. Beneficially, in contrast to PCR alone, 'SNAILS' biosensor assays can potentially provide single base resolution detection. Although crucial for the 'SNAILS' assay, the PCR amplification step is also useful in that it could also enable for positive detection at sites of low parasitic levels, that might be undetected by other methods without an amplification step.

We envision our 'SNAILS' biosensors as a complementary addition to the current schistosomiasis detection toolbox, in that it can provide species differentiation data which can highlight samples of interest for further analysis, such as full genomic sequencing. Likewise, bioinformatic insights from genomic sequencing studies can also be used to further refine or increase the number of validated 'SNAILS' biosensor probes—thus enhancing species-specific detection capabilities. Genomic sequencing and clinical validation of schistosome drug resistance markers, along with new workflows for integrating patient or veterinary samples (e.g. urine) into 'SNAILS' biosensor workflows, might also enable future clinical or agricultural point-of-care applications. In conclusion, we believe that the inclusion of 'SNAILS' biosensor assays may, in the future, enable epidemiological screening of test sites for basic research, environmental monitoring and WASH interventions, as well as monitoring the effectiveness of such interventions at these sites.

## Methods

### Identification of schistosome targets and probe design

*S. mansoni* AJ519524 [56] and MG562513 (GenBank), *S. rodhaini* AY157202 [57], *S. haematobium* GU257336 and GU257338 [35], *S. bovis* MH647124 [58] and FJ897160 [59], *S. guineensis* AJ519517, AJ519522 and AJ519223 [56], *S. curassoni* AJ519516 [56] and AY157210 [57], *S. japonicum* EU325878 (GenBank), *S. malayensis* EF635956 [60] and *S. mekongi* EF635955[60] *cox*1 nucleotide sequences were obtained from GenBank and aligned using MUSCLE [47] with default parameters (S1A Fig). Resultant schistosome targets identified and probe designs are listed in S1A Table. Oligonucleotide primers used as synthetic biosensor targets, for cloning and for sequencing, as well as those comprising the biosensor probes are listed in S1A Table and were ordered from Integrated DNA Technologies (IDT, USA).

### Adult *Schistosoma* worm samples

*S. mansoni* adult worms 1 and 2 (SM_AW1 and SM_AW2), *S. mansoni* adult worm isolates 2740 (SM_C; isolated from Kenya), 3112 (SM_D; isolated from Senegal), 4209 (SM_E; isolated from Oman), N1g274 (SM_F; isolated from Nigeria), 4664–2 and 4464–4 (SM_G and SM_H; both isolated from Uganda) and *S. haematobium* adult worms 1 and 2 (SH_AW1 and SH_AW2) were obtained from the Schistosomiasis collection at the Natural History Museum (SCAN) [61].

### Purification and elution of *Schistosoma* worm gDNA

*Schistosoma* adult worm gDNA was obtained using the DNeasy blood and tissue kit (catalogue #69504; Qiagen, UK) following the manufacturer's instructions, with the purified gDNA eluted into 100 μl nuclease-free water (nH$_2$O).

### Amplification of parasite gDNA

Polymerase chain reactions (PCR) were used to amplify both the longer *cox*1 gene fragment and the 180-base target regions. For the longer gene fragment, gDNA from adult worms were amplified using Q5 high-fidelity DNA polymerase (catalogue #M0491S; New England Biolabs, USA) following the manufacturer's instructions for 50 μl reactions. Essentially, 1 μl of gDNA sample was amplified using 2.5 μl (10 μM working stock) each ofprimers Cox1_schist_5' (AJW485) and Cox1_schist_3' (AJW486, 57] in a ProFlex PCR System (Applied Biosystems, Thermo Fisher Scientific, USA) with the following protocol: 1 cycle of 98˚C for 30 seconds, 30 cycles of 98˚C for 10 seconds, 57˚C for 30 seconds and 72˚C for 90 seconds, followed by a final cycle of 72˚C for 2 minutes. For the 180-base target region, 1 μl of gDNA (for all adult worm samples) or 1 μl of plasmid DNA (10 ng DNA total) were amplified using primer pairs 5-SM-PCR/3-SM-PCR (AJW870/AJW871; *S. mansoni*-specific) or 5-SH-PCR/3-SH-PCR (AJW872/AJW873; *S. haematobium*-specific) in a ProFlex PCR System (Applied Biosystems, Thermo Fisher Scientific, MA, USA) with the following protocol: 1 cycle of 98˚C for 30 seconds, 40 cycles of 98˚C for 10 seconds, 57˚C for 30 seconds and 72˚C for 30 seconds, followed by a final cycle of 72˚C for 2 minutes.

DNA samples to be cloned into the pCR-Blunt or pCR-Blunt II-TOPO plasmids were purified using the DNA clean & concentrator kit (catalogue #D4033; Zymo Research, USA). Visualisation for the validation of the 180-base target PCR products was undertaken using agarose gel electrophoresis: 5 μl of PCR product was separated using a 1.5% agarose gel with SYBR safe DNA stain (catalogue #S33102; Invitrogen, Thermo Fisher Scientific, MA, USA) in 1X Tris-acetate-EDTA buffer for 30 minutes at 100 volts. Quick-load 1 kb plus DNA ladder (100 bp-10

kb; catalogue #N0469S; New England Biolabs, USA) was also loaded onto the agarose gel to enable sizing of the PCR product. Bands on the agarose gel were visualised using the Gel Doc XR+ system with Image Lab software (model # Universal Hood II; Bio-Rad Laboratories Inc., USA).

## Cloning of PCR products into pCR-Blunt or pCR-Blunt II-TOPO

PCR products amplified were cloned into either pCR-Blunt or pCR-Blunt II-TOPO plasmids following the manufacturer's instructions (catalogue #K270020 & #450245; Invitrogen, Thermo Fisher Scientific, MA, USA). Reactions were transformed into *Escherichia coli* NEB10-beta or NEB5-alpha. For plasmid recovery, *E. coli* strains were grown in Luria-Bertani (LB) medium supplemented with 35 μg/ml Kanamycin and cultured at 37˚C with shaking (220 rpm). The plasmid DNA from these cultures were purified using the QIAprep spin mini-prep kit as per the manufacturer's instructions (catalogue #27106; Qiagen, UK), with samples eluted in 50 μl of nH$_2$O. The DNA sequences of the inserts were verified by the sequencing service provided by Eurofins Genomics GmbH (Ebersberg, Germany). The resulting plasmids are detailed in S1C Table.

## Generation of ssDNA

After validation by agarose gel electrophoresis, PCR reactions for the 180-base target regions were enzymatically treated with Lambda Exonuclease (catalogue# EN0562; Thermo Fisher Scientific, MA, USA) to generate ssDNA. To the 45 μl PCR reactions 1 μl Lambda exonuclease (10 U/μl), 6 μl 10X reaction buffer (670 mM glycine-KOH (pH 9.4), 25 mM MgCl$_2$, 0.1% (v/v) Triton X-100) and 8 μl nH$_2$O were added to give a total reaction volume of 60 μl. The reactions were mixed gently via pipetting and incubated at 37˚C for 30 minutes, 80˚C for 10 minutes to inactivate the lambda exonuclease and then 10˚C for 5 minutes in a ProFlex PCR System (Applied Biosystems, Thermo Fisher Scientific, MA, USA).

The ssDNA produced was subsequently purified using the ssDNA/RNA clean & concentrator kit (catalogue #D7011; Zymo Research, USA) as per the manufacturer's instructions, with ssDNA samples being eluted in 12 μl nH$_2$O. The concentration of the purified ssDNA samples was determined, according to the manufacturer's instructions, using a Qubit 3 Fluorometer (Thermo Fisher Scientific, MA, USA) and a Qubit ssDNA assay kit (catalogue #Q10212; Thermo Fisher Scientific, MA, USA).

## SNAILS assay

The SNAILS assay developed in this study is adapted from a spinach-based fluorescent light-up assay for microRNAs [40]. However, in this study we use ssDNA as a target. The assay has four steps: preparation of the probes and ssDNA targets, annealing of probes and targets, ligation of the two half probes, aptamer transcription and fluorescence detection of the aptamer.

## Preparation of the probes, ssDNA targets and DFHBI-1T fluorogen

Probe DNA stock solutions (100 μM in nH$_2$O) were diluted in nuclease-free duplex buffer (30 mM HEPES, pH 7.5; 100 mM potassium acetate (CH$_3$CO$_2$K); catalogue #11-01-03-01, IDT, USA) to the required concentration for the reaction tested. The ssDNA target stock solutions (100 μM in nH$_2$O) were diluted in nH$_2$O to the required concentration for the reaction tested. The '2$^{nd}$ T7 promoter sequence' oligonucleotide (AJW694) stock solution (100 μM in nH$_2$O) was diluted 1 in 100 in nuclease-free duplex buffer (30 mM HEPES, pH 7.5; 100 mM CH$_3$CO$_2$K; catalogue #11-01-03-01, IDT, USA) to give a 1 μM working solution. DFHBI-1T

fluorogen (catalogue #5610/10; R & D systems, Bio-Techne, MN, USA) stock solution (100 mM in dimethyl sulphoxide (DMSO)) was diluted 1 in 200 in $nH_2O$ to give a 500 μM working solution.

## Annealing of probes and target ssDNA

Annealing reactions (7 μl) comprised 1 μl probe half A, 1 μl probe half B, 1 μl target DNA (or 1 μl $nH_2O$ for negative controls) and 4 μl nuclease-free duplex buffer (30 mM HEPES, pH 7.5; 100 mM $CH_3CO_2K$; catalogue #11-01-03-01, IDT, USA) and mixed gently via pipetting. Concentrations of probes and target ssDNA added are based on the 10 μl volume of the ligation reaction. For example, for reactions requiring 200 nM of each half probe in the final 10 μl ligation reaction, 1 μl of 2 μM probe DNA in nuclease-free duplex buffer (30 mM HEPES, pH 7.5; 100 mM $CH_3CO_2K$; catalogue #11-01-03-01, IDT, MA, USA) is added to the annealing reaction. Likewise, for reactions requiring 10 μM of target ssDNA in the final 10 μl ligation reaction, 1 μl of 100 μM target ssDNA in $nH_2O$ is added to the annealing reaction. For reactions using ssDNA samples generated from PCR reactions or the equivalent control oligonucleotide, 1 μl of the required sample amount (ng/μl) was added. Annealing reactions were incubated at 95°C for 3 minutes in a ProFlex PCR System (Applied Biosystems, Thermo Fisher Scientific, USA) and then incubated on ice for 5 minutes.

## Ligation of probes

To the 7 μl annealing reactions, 1 μl $nH_2O$, 1 μl T4 DNA ligase (400 U; catalogue #M0202L; New England Biolabs) and 1 μl 10X T4 DNA ligase reaction buffer (final 1X concentration in reaction: 50 mM Tris-HCL, 10 mM $MgCl_2$, 1 mM ATP, 10 mM DTT, pH 7.5) were added and reactions mixed gently via pipetting. Ligation reactions were incubated at 25°C for 20 minutes, 65°C for 10 minutes to inactivate the T4 DNA ligase and then 25°C for 5 minutes in a ProFlex PCR System (Applied Biosystems, Thermo Fisher Scientific, USA).

## T7 transcription and real-time fluorescence detection

T7 transcription reactions (60 μl final volume) using the TranscriptAid T7 high yield transcription kit (catalogue #K0441; Thermo Scientific, Thermo Fisher Scientific, USA), were set up as follows: 1.5 μl RNase Inhibitor (60 U; catalogue #M0314L; New England Biolabs), 24 μl NTP mix (final concentration of 10 mM each of ATP, CTP, GTP and UTP), 12 μl 5X TranscriptAid reaction buffer (catalogue #K0441; Thermo Scientific, Thermo Fisher Scientific, USA), 6 μl dithiothreitol (DTT; 0.1 M solution; catalogue #707265ML; Thermo Scientific, Thermo Fisher Scientific, USA), 6 μl TranscriptAid enzyme mix (catalogue #K0441; Thermo Scientific, Thermo Fisher Scientific, USA), 3 μl 2nd T7 promoter sequence oligonucleotide (1 μM stock solution), 6.3 μl of the ligation product, and 1.2 μl DFHBI-1T (10 μM final concentration). Reactions were mixed gently via pipetting. Three 15 μl aliquots of each reaction were loaded into 384-well plates (catalogue #781096, Greiner Bio-One, Kremsmünster, Austria) and measured using a CLARIOstar plate reader (BMG, Ortenberg, Germany) with the following settings: Excitation 440–15 nm/Emission 510–20 nm, orbital shaking for 5 seconds at 500 rpm before measurement every 5 minutes. Delta (Δ) fluorescence measurements (a.u.) per hour were calculated between 20 and 80 minutes of the reactions.

## Statistics

Statistical analysis (standard error of the mean (s.e.m.)) was carried out using GraphPad Prism 9.1.1 (GraphPad Software Inc., La Jolla, California).

## Supporting information

**S1 Text. Supplementary figures, data and tables.**
(DOCX)

**S1 Data. Excel spreadsheet.** Excel spreadsheet showing biosensor assay data. The label on the spreadsheet tabs indicate the corresponding figure in the main text.
(XLSX)

## Acknowledgments

We would like to thank our WISER project colleagues for their expertise and supportive comments. We also thank colleagues in the Section of Structural and Synthetic Biology within the Department for Infectious Disease at Imperial College London for their advice and helpful comments.

## Author Contributions

**Conceptualization:** Alexander James Webb, Fiona Allan, Richard J. R. Kelwick, Feleke Zewge Beshah, Safari Methusela Kinung'hi, Michael R. Templeton, Aidan Mark Emery, Paul S. Freemont.

**Data curation:** Alexander James Webb, Fiona Allan, Richard J. R. Kelwick.

**Formal analysis:** Alexander James Webb, Fiona Allan.

**Funding acquisition:** Feleke Zewge Beshah, Safari Methusela Kinung'hi, Michael R. Templeton, Aidan Mark Emery, Paul S. Freemont.

**Investigation:** Alexander James Webb, Fiona Allan, Richard J. R. Kelwick.

**Methodology:** Alexander James Webb, Fiona Allan, Richard J. R. Kelwick, Aidan Mark Emery.

**Project administration:** Alexander James Webb, Michael R. Templeton, Paul S. Freemont.

**Resources:** Fiona Allan, Michael R. Templeton, Aidan Mark Emery, Paul S. Freemont.

**Supervision:** Michael R. Templeton, Paul S. Freemont.

**Validation:** Alexander James Webb, Richard J. R. Kelwick.

**Visualization:** Alexander James Webb.

**Writing – original draft:** Alexander James Webb, Fiona Allan, Richard J. R. Kelwick.

**Writing – review & editing:** Alexander James Webb, Fiona Allan, Richard J. R. Kelwick, Feleke Zewge Beshah, Safari Methusela Kinung'hi, Michael R. Templeton, Aidan Mark Emery, Paul S. Freemont.

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
