## [Decision Letter · Decision Letter 0]

9 Mar 2022

Dear Prof. Freemont,

Thank you very much for submitting your manuscript "Specific Nucleic AcId Ligation for the detection of Schistosomes: SNAILS" for consideration at PLOS Neglected Tropical Diseases. As with all papers reviewed by the journal, your manuscript was reviewed by members of the editorial board and by several independent reviewers. In light of the reviews (below this email), we would like to invite the resubmission of a significantly-revised version that takes into account the reviewers' comments. 

We cannot make any decision about publication until we have seen the revised manuscript and your response to the reviewers' comments. Your revised manuscript is also likely to be sent to reviewers for further evaluation.

Sincerely,

Neil D. Young

Associate Editor

Walderez Dutra

Deputy Editor

Reviewer's Responses to Questions

**Key Review Criteria Required for Acceptance?**

**Methods**

-Are the objectives of the study clearly articulated with a clear testable hypothesis stated?

-Is the study design appropriate to address the stated objectives?

-Is the population clearly described and appropriate for the hypothesis being tested?

-Is the sample size sufficient to ensure adequate power to address the hypothesis being tested?

-Were correct statistical analysis used to support conclusions?

-Are there concerns about ethical or regulatory requirements being met?

Reviewer #1: The objective of the study is to demonstrate proof-of-concept for a DNA-based biosensor for species-specific detection of i) synthetic DNA targets; and ii) field-collected samples, from two species of schistosome that cause human health problems. This is testable and the study design is appropriate to address the stated objectives for the synthetic targets derived from reference sequences as the authors use a total of nine closely related species (some with multiple sequenced genomes) to ensure specificity and they go on to extensively optimise a number of relevant parameters.

However, the use of only two adult worms of each species collected in the field is a concern as one of the two S. haematobium already had two base changes in a probe target region. There is no discussion of the potential or measured rate of variation in nature at those key base positions. They artificially mutated a range of single, double or multiple base changes to the S. mansoni wild-type target region to demonstrate that the probes retained specificity but this strand could be tied more clearly to robustness against likely natural variation.

It is also unclear why DNA from only two organisms for each species was available for the study as this is not a large sample size. Without further understanding of natural sequence variation it is difficult to conclude that the probes would be specific in a large scale study of samples from nature so I would expect to see this addressed in the discussion. 

The statistical methods deployed are appropriate to support the conclusions. There are no concerns around ethical or regulatory requirements.

**Results**

-Does the analysis presented match the analysis plan?

-Are the results clearly and completely presented?

-Are the figures (Tables, Images) of sufficient quality for clarity?

Reviewer #1: The analysis presented matched the analysis plan and results are clearly and completely presented but may be too complete in that some of the information in the captions is easily discoverable in the methods so need not be repeated in each figure.

For example, most figures contain “Fluorescence outputs of reactions were analysed using a CLARIOstar plate reader (440-15 nm/510-20 nm; Gain 1500).” This information is already presented in the methods and could be removed from the captions. Likewise, “Mean and Standard error of the mean calculated using GraphPad Prism 9.1.1.” In the supplementary table captions is redundant to the methods section.

I request that the authors include a reference to the supplementary information in Fig 2,3,5 and 6 captions where their probe sequence data is included as a non-copyable image, so the reader knows that they can find a text version elsewhere. 

Supplementary Figure 9, 10, 16, 17 are not high resolution in the PDF provided.

The PDF contains a very large number of tables with hourly plate reader date and calculations. This makes it difficult to find the other information that the reader is more likely to need, like the tables of nucleic acid sequences. Given this and that PLOS recommends that spreadsheets are preferable to PDFs or images when providing tabulated data, I would suggest providing a spreadsheet containing all of this experimental data (e.g. one experiment per sheet) rather than a PDF table, leaving the PDF shorter and containing only figures and resource tables.

**Conclusions**

-Are the conclusions supported by the data presented?

-Are the limitations of analysis clearly described?

-Do the authors discuss how these data can be helpful to advance our understanding of the topic under study?

-Is public health relevance addressed?

Reviewer #1: The conclusion is that the biosensor can detect and differentiate between two human health relevant schistosomes, S. mansoni and S. haematobium, as well as discriminate these from other closely related species such as S. bovis and S. rodhaini and that ‘SNAILS’ probes are amenable to rapid design iterations. This is supported by the data presented but I would further discuss the very limited sample size for field-collected samples and describe the level of intraspecies sequence variation that exists (if known) or may potentially exist, given one of the four samples tested had two base pairs variance. The rapid design iterations will mitigate against this causing issues during the design phase but not in field use if the natural population has high variance – for this I would expect to see a much larger sample size. The public health relevance of the study is described in terms of being an environmental surveillance tool to monitor presence of the schistosomes and the effectiveness of control measures.

I would also encourage some discussion of future developments which would be needed to advance the biosensor design as the introduction states that the authors concentrated "on the need for a species-specific point-of-care (PoC)/point-of-test (PoT) diagnostic" but there is then no further mention of the PoT potential for this assay, which currently uses PCR, one of the harder amplification methods to convert to PoT, followed by a further enzymatic step and then addition of the probe and ligase. It is also unclear if the fluorescent output is easy to read without a real-time system. These could be limitations for a PoT approach.

SNAILS is a redesign of the spinach aptamer assay for ssDNA detection and applications of this to other targets could be mentioned in the discussion alongside any planned further research.

**Editorial and Data Presentation Modifications?**

Reviewer #1: Lines 207-250 belong in the introduction as they provide further detail about existing control measures and the landscape of existing molecular diagnostics into which this project fits, adding context and situating it within the literature. I would include content from line 250 onwards as discussion and add some further detail on natural sequence variation in populations and its potential effect on the assay specificity.

L.244 the comment that sequencing is expensive should be put in context and prefaced by a comment that the previous techniques would typically need to be followed by sequencing to enable species level identification.

There are a number of typos in the text but they all look like they should be detected by spell check so I haven’t listed them.

**Summary and General Comments**

Reviewer #1: Overall this is a good lab-level proof of concept for a diagnostic biosensor design with good potential for impacting surveillance efforts in schistosomiasis. The design is a novel and original adaptation of an existing assay to detect ssDNA instead of RNA which may be of interest to other biosensor designers. It is intended to eventually be PoT but the proof of concept was performed with standard lab instrumentation and multiple steps in the lab with no further information provided on PoT potential, which makes it less relevant in its current form to a wide audience of practitioners. 

Being species-specific without the need for sequencing is a compelling alternative to existing assays that require sequencing to provide that level of specificity. The major weakness of the study is the aforementioned small sample size for organismal DNA and lack of clarity on intraspecies sequence variation in the field which means it falls short of demonstrating broader application. However, this is consistent with the claims of the authors, who clearly describe it as a proof of concept. The experiments appear to have been well executed and data is provided in full. I have no further comments on the data presented and its analysis and interpretation.

Demonstrating successful detection across a much wider set of field samples would improve the paper's importance and relevance to researchers, practitioners, or policy makers in the field of NTDs. This could involve further collection of field samples if these are not already available to the authors, which could take significant amounts of time. The paper stands alone in terms of the biosensor design so could be published in another journal with different criteria with quite minor revisions, particularly recognising limitations in the discussion section, but I feel this major revision would substantially improve its fit with the PLOS NTD criteria. 

An exception would be if the authors can convincingly demonstrate without additional experimental work (e.g. through presentation of secondary data from schistosome genetic/genomic studies or other means) that the likelihood of sequence mutations across a broad population within each species is low enough that the lab results obtained with synthetic DNA from reference genomes is representative of likely performance in the field. It is possible that this data exists but was not presented clearly in the paper. They did artificially mutate a range of single, double or multiple base changes to the S. mansoni wild-type target region to demonstrate specificity and may consider this is sufficient evidence that natural variation will not perturb the function of the assay. Discussing this in more detail and why it is likely a good proxy for natural variation could also be sufficient.

PLOS authors have the option to publish the peer review history of their article (what does this mean?). If published, this will include your full peer review and any attached files.

Reviewer #1: No
---

## [Decision Letter · Decision Letter 1]

4 Jul 2022

Dear Prof. Freemont,

We are pleased to inform you that your manuscript 'Specific Nucleic AcId Ligation for the detection of Schistosomes: SNAILS' has been provisionally accepted for publication in PLOS Neglected Tropical Diseases.

Best regards,

Neil D. Young

Associate Editor

Walderez Dutra

Deputy Editor

Reviewer's Responses to Questions

**Key Review Criteria Required for Acceptance?**

**Methods**

-Are the objectives of the study clearly articulated with a clear testable hypothesis stated?

-Is the study design appropriate to address the stated objectives?

-Is the population clearly described and appropriate for the hypothesis being tested?

-Is the sample size sufficient to ensure adequate power to address the hypothesis being tested?

-Were correct statistical analysis used to support conclusions?

-Are there concerns about ethical or regulatory requirements being met?

Reviewer #2: I have reviewed the revisions made by the authors in response to Reviewer #1’s comments. I am in agreement with the issues raised by Reviewer #1 and I would have raised the same issues regarding the small sample size. The authors have carried out additional bioinformatic analyses on a range of publicly available datasets. I am satisfied that these additional analyses address the concerns of Reviewer #1.

**Results**

-Does the analysis presented match the analysis plan?

-Are the results clearly and completely presented?

-Are the figures (Tables, Images) of sufficient quality for clarity?

Reviewer #2: Yes, The updated images have been improved as suggested by Reviewer #1

**Conclusions**

-Are the conclusions supported by the data presented?

-Are the limitations of analysis clearly described?

-Do the authors discuss how these data can be helpful to advance our understanding of the topic under study?

-Is public health relevance addressed?

Reviewer #2: Yes, these factors has been addressed

**Editorial and Data Presentation Modifications?**

Reviewer #2: (No Response)

**Summary and General Comments**

Reviewer #2: (No Response)

PLOS authors have the option to publish the peer review history of their article (what does this mean?). If published, this will include your full peer review and any attached files.

Reviewer #2: No

---

## [Editor Report · Acceptance letter]

19 Jul 2022

Dear Prof. Freemont,

We are delighted to inform you that your manuscript, "Specific Nucleic AcId Ligation for the detection of Schistosomes: SNAILS," has been formally accepted for publication in PLOS Neglected Tropical Diseases.

Best regards,

Shaden Kamhawi

co-Editor-in-Chief

Paul Brindley

co-Editor-in-Chief
